# Canine-Origin Platelet-Rich Fibrin as an Effective Biomaterial for Wound Healing in Domestic Cats: A Preliminary Study

**DOI:** 10.3390/vetsci8100213

**Published:** 2021-09-30

**Authors:** Carla S. Soares, Isabel R. Dias, Maria A. Pires, Pedro P. Carvalho

**Affiliations:** 1Laboratory of Histology and Anatomical Pathology, University of Trás-os-Montes e Alto Douro (UTAD), 5000-801 Vila Real, Portugal; carlasoares.medvet@gmail.com (C.S.S.); apires@utad.pt (M.A.P.); 2Department of Veterinary Sciences, School of Agricultural and Veterinary Sciences (ECAV), University of Trás-os-Montes e Alto Douro (UTAD), 5000-801 Vila Real, Portugal; idias@utad.pt; 3CECAV—Centre for Animal Sciences and Veterinary Studies, UTAD, 5000-801 Vila Real, Portugal; 4CIVG—Vasco da Gama Research Center, University School Vasco da Gama—EUVG, Av. José R. Sousa Fernandes, Campus Universitário, Lordemão, 3020-210 Coimbra, Portugal; 5Vetherapy-Research and Development in Biotechnology, 3020-210 Coimbra, Portugal

**Keywords:** platelet-rich fibrin, xenogenic, platelet therapy, wound healing, regenerative medicine, cat

## Abstract

Platelet-rich fibrin (PRF) is a recent platelet-based biomaterial, poised as an innovative regenerative strategy for the treatment of wounds from different etiologies. PRF is defined as a biodegradable scaffold containing elevated amounts of platelets and leukocytes having the capability to release high concentrations of bioactive structural proteins and acting as a temporal release healing hemoderivative. This study aimed to evaluate the performance of canine-origin PRF, obtained from blood of screened donors, as a regenerative biomaterial suitable for the treatment of critical wounds in felines. Four short-hair felines with naturally occurring wounds were enrolled in this study. Three of the wounds were considered infected. Each PRF treatment was the result of the grafting of newly produced PRFs at the recipient area. The PRF treatment was initially performed two to three times per week, followed by single weekly treatments. The study was finalized when complete wound closure was achieved. No topical antimicrobial/antiseptic treatment was applied. The present research demonstrated that xenogenic PRFs significantly induced healthy vascularized granulation tissue in lesions with soft tissue deficit, also prompting the epithelization at the injured site. No rejection, necrosis, or infection signs were recorded. Additionally, PRF-therapy was revealed to be a biological cost-effective treatment, accelerating the wound healing process.

## 1. Introduction

Platelet-rich fibrin (PRF) is a versatile platelet-enriched product available for veterinary regenerative treatments, initially developed for human oral and maxillofacial surgery [1,2]. PRF is a recent platelet-based biomaterial, poised as an innovative regenerative strategy for the treatment of wounds from different etiologies [3,4,5,6]. This biomaterial is attained from feasible small-volume blood samples, collected through the venous vessels and subsequent centrifugation, without the requirement of additive agents to promote the clot, expensive equipment, or advanced operator skills [1,4,7,8]. The PRF, also recognized as leukocyte-rich-PRF [9], is defined as a natural fibrin matrix containing elevated amounts of platelets and leukocytes, being presented as an organic and biodegradable scaffold having the capability to release high concentrations of bioactive structural proteins and acting as a temporal release healing promotor in tissue angiogenesis and remodeling [2]. Studies focused on the temporal profile of the PRFs’ secretome have shown that the active secretome released from the clots is maintained even after its application at wound bedding [2,10]. Several growth factors (GFs), such as platelet-derived growth factor-BB (PDGF-BB), transforming growth factor beta-1 (TGF-β1), and vascular endothelial growth factor-A (VEGF-A), as well other important cytokines, are released after platelet and/or leukocytes degranulation, initiated during the centrifugation process [11]. The success of PRF in the human clinical context has been supported by both in vitro and in vivo scientific research [2,12,13,14,15]. The increasing interest in platelet-based therapies has contributed to the development of novel veterinary treatments [16,17]. A recent in vitro study conducted by the authors of the present work also demonstrated the burst of PDGF-BB, TGF-ß1, VEGF-A, and interleukin-8 concentration released from both canine and feline intact PRFs over 10 days [18]. Moreover, in clinical settings, PRFs produced from felines may prove to be a limited strategy, since domestic cats are considered low blood-volume donors and, in some cases, non-cooperating individuals. The continuous blood collections necessary for the clot production acquired for the treatment until its finalization (the complete wound closure) constitutes a major constraint, considering feline-PRF hemoderivatives. Anticipating these limitations, xenogenic administration of PRFs may be a more suitable option, considering that topical use of xenogenic platelet-based formulations have been recently reported in the veterinary literature [19,20].

This study aimed to investigate the therapeutic safety and efficacy of xenogenic PRFs, also designated as heterologous, derived from screened canine donors, for the treatment of naturally occurring skin wounds in compromised feline patients, where the obtention of autologous PRF is not feasible.

## 2. Materials and Methods

### 2.1. Production of Xenogenic PRF Clots

The present study was approved by the Ethics Committee of the University of Trás-os-Montes-e-Alto-Douro (UTAD, 60-CE-UTAD-2020) and with written informed consent by the animals’ owners.

Four dogs (1 female and 3 males), aged between 2 to 8 years, clinically and hematologically healthy, were selected as PRF donors (Table 1). These animals were tested negative for *Leishmania infantum* and *Dirofilaria immitis*, had a regularly updated vaccine and deworming schedule, and lived mostly indoors and without a traveling record. Each PRF clot was produced as described in previous works [16,18]. Briefly, after local trichotomy of the area and skin cleansing, 5 mL of whole blood was collected to sterilized conical base polypropylene tubes (57 × 15.3 mm), without clot activator. The centrifugation was immediately performed after blood collection, at 580× *g* (3000 rpm) for 10 min and was performed using an in-clinics 45° angle rotor centrifuge (Orto Alresa^®^ RT 114, NS 080214/02, ø 8.2 cm, Madrid, Spain) at room temperature. After centrifugation, the blood rested inside the tube for 60 min. The PRFs were then removed from the tubes, and the red fraction was discarded using sterile tweezers (Figure 1).

### 2.2. Characterization of the Recipient Feline Population

Four short-hair domestic cats (1 female and 3 males), aged between 9 months to 10 years, were enrolled in this study. From the total of four wounds, three (3/4) were considered infected (cases 2, 3, and 4).

Two (2/4) felines were positive for feline immunodeficiency virus (FIV) and presented moderate anemia at the consultation. One (1/4) of these patients was positive for *Bartonella* sp. (diagnosis made by polymerase chain reaction assay).

The clinical characterization of the recipients is summarized in Table 2.

### 2.3. PRF Grafting Procedure and Treatment Protocol

Lesions were irrigated with sterile saline solution before each new treatment to remove all exudate. Wound debridement was only performed on day 1, if required. The first day of treatment was designated as day one (D1). One PRF clot was applied for each 1.5 to 2 cm^2^ wound area, ensuring that all the wounds were completely covered by the applied PRFs. Each PRF treatment was constituted by the application of newly processed PRFs at the recipient area, assuring the contact between the PRFs platelet-rich area and the lesion. After the PRF application, a closed bandage was applied using a sterile paraffin gauze bandage maintaining the clot at the recipient area.

The PRF treatment was initially performed two to three times per week in order to increase the in situ concentration of cytokines and GFs at the wound site when the dimension of the lesion was higher and also generally wet (initial exudative phase of wound progression). Single treatments were applied from the second week until the wound epithelization was achieved. The application of PRFs was suspended if exophytic granulation tissue was observed in noteworthy contracted wounds. The bandaging procedure was continuously performed until complete wound closure was reached. All the patients received conventional systemic antimicrobial and non-steroidal anti-inflammatory treatment (see Appendix A).

### 2.4. Assessment of the Wound Area Reduction and Statistical Analysis

The lesion was assessed by measuring the wound size using a metric tape at each time point (Table 3). The wound depth was not considered. All the patients (with the exception of case 1) were observed 6 months after the wound closure.

The wound area during the healing progress was assessed by the same researcher using ImageJ^®^ software (version Image J: 2.1.0/1.53c, Bethesda, MD, USA). Results are expressed as the median, minimum, and maximum values. Statistical analysis was conducted using Prism Version 6 (GraphPad^®^ Software Inc., La Jolla, CA, USA). The percentage of wound contraction (%WC) was calculated using the following formula: [(Wound Area at day 1−Wound Area at Specific Timepoint)/Initial Wound Area] × 100.

## 3. Results

### 3.1. Assessment of the Wound Area over Time

The wound area of each lesion documented throughout time is represented in Figure 2. In case 1, the PRFs treatment was not conducted until wound closure, and a hind limb amputation was the clinical solution considered by the owners.

The initial surface area of the major skin wounds recorded 5.47 cm^2^ (case 1) and 7.01 cm^2^ (case 4), receiving 2 and 4 PRFs, respectively, providing complete wound covering. Cases 2 and 3 required 1 PRF, recording, respectively, a surface area of 2.26 and 3.42 cm^2^.

Regarding case 1, measurement of the lesion was not performed at the second PRF grafting (day 3). At the moment of the second PRF-grafting (day 3), the three wounds (cases 2, 3, and 4) ranged from 0.64 cm^2^ to 5.68 cm^2^ (median 2.28 cm^2^), recording 0.64 cm^2^, 2.28 cm^2^, and 5.68 cm^2^, respectively.

On average in the four cases, the four wounds recorded a median reduction of 59.98% (30.00–97.06) after the second PRF-treatment between days 5 and 9. Case 1 did not proceed with the study after day 8 (Figure 3), as previously described. No topical antimicrobial/antiseptic treatment was applied. No rejection, necrosis, or infection signs were recorded in any wound.

Regarding the three cases that completed the study, the average time for complete wound healing varied according to the lesion size, occurring between days 11 and 42. At the second week of treatment, complete wound healing was achieved in cases 2 and 3.

### 3.2. Wound Healing Progress during the PRF Treatment

In all the cases, PRF treatments were well tolerated, revealing a significant induction of granulation tissue, especially observed in the largest lesions, as documented by photographic records. No infection signs were recorded. Nevertheless, the presence of exudate was documented, especially in two of the wounds in the first week of treatment (cases 2 and 3; Figure 4 and Figure 5, respectively), decreasing significantly with time. Fibrinous material deposition was not observed in most of the cases, being minimal in the largest lesion (case 4; Figure 6).

Each PRF treatment induced a remarkable granulation tissue at the wound bedding in all patients. No signs of inflammation or infection were observed in the peripheric area and/or at wound site.

Canine PRFs treatments did not reveal any adverse effects in the felines, not even in the animals (1 and 4) that received clots from two different donors at different time points.

Wound contraction (Figure 7), re-epithelialization, and crust formation were progressively witnessed in all the three cases. Ischemic or necrotic tissue was not observed in any of the cases under any circumstance. Moreover, intense vascularization of the wound bed was observed by the presence of red exophytic granulation tissue, easily friable and bleeding to touch, supporting the advanced in situ tissue augmentation, and therefore, the healing progress.

All the treatments resulted in healthy tissue, forming vestigial aesthetic scars. Evident skin contraction was recorded in the larger injury (case 4). There was no wound recurrence up to 6 months after PRF therapy and wound closure.

## 4. Discussion

PRF therapy in the human field has been recently studied and reported in a few clinical studies for the treatment of specific wounds, such as those resulting from diabetes and burns, demonstrating positive results [21,22,23,24]. The growing evidence reported by several different studies strongly supports the therapeutic use of platelet-based products, defined as biocompatible and biomimetic agents—namely, due to their anti-inflammatory, pro-healing, and even analgesic properties [25,26]. The determinant role of platelets in the wound healing process is currently accepted [27], also accepted is that the natural scaffold formed by the fibrin arrangement plays an important part in the healing processes through the promotion of physiologic neoangiogenesis [28,29].The concentration of leukocytes observed within the PRF clot also directly supports the tissue remodeling and regeneration [30]. Lymphocytes are considered local regulators and effector-cells during healing, demonstrated by the use of PRF membranes in human dentistry, releasing increased quantities of GFs and cytokines over a long period [31,32,33]. The easy PRF’s obtention procedure (a safe, easy, and cost-effective methodology, with no equipment or requirements of advanced manufacturing operator skills), associated with an inherent low number of variables in its preparation and significantly minimal risk of secondary contamination associated with its manipulation, has contributed to the broader application of this biological product [17].

The PRF clots applied as a grafting material in this research were prepared without the addition of clotting agents, which were recently debated and considered not safe for clinical use [34]. Generally, the natural polymerization of the clots occurred within 60 min after the centrifugation step, and a recent study documented the stability of PRF clots over this period, with the increase in platelet concentration during these 60 min periods [35].

To date, PRF usage in veterinary applications has been rarely described, despite the promising results accomplished in human clinical trials [19,20]. PRF preparation methodology is easily reproducible, recording consistent clinical performance, particularly in platelet concentration and related GFs, cytokines, and hormones [36].

Historically, the use of autologous hemoderivatives has been more widely reported, but the clinical application of allogenic hemoderivatives, such as blood transfusions, has also been currently used in both human and veterinary clinical settings [37,38]. More recently, topical use of xenogenic platelet-based formulations has been described: a neutered domestic short-haired cat presenting a large skin traumatic wound was successfully treated with canine platelet-rich plasma (PRP). The wound, initially contaminated, healed completely within 20 days with no adverse reactions recorded [19]. Additionally, an adult domestic short-hair cat presenting an infected corneal ulceration as a consequence of an ocular traumatic injury was treated with canine-derived PRP eye drops, instilled twice daily for 1 month. No adverse reactions were recorded. The recovery of the cornea was observed with a corneal leukoma remaining in the central area [20]. Nevertheless, the application of xenogenic PRF in veterinary treatments is prone to the latent risk of infectious agents’ transmission (particularly misidentified in periods of subclinical disease), and is even prone to induce antigenic-immune reactions in the recipient [35].

Felines are naturally small blood-volume donors, limiting the autologous administration of platelet-based formulations, especially if we consider cats that have pre-existing hematological, immunological, and/or metabolic disorders. In this study, two felines were FIV positive, one was also infected with *Bartonella* sp., and a third case developed hepatic lipidosis secondary to the trauma. In three cases, the presence of anemia was the major distress factor for blood harvestings. The data gathered during this study point to xenogenic cutaneous PRF-graphs as a suitable strategy for feline patients, considering the limitations of cats as limited blood-volume donors and especially contemplating cases where the existence of co-morbidities does not allow the utilization of autologous PRFs. Moreover, principles related to the prevention of immunogenic reactions are always considered in grafting techniques involving hemoderivatives, which is supported by the election of the lowest immunogenic blood type source. This research did not observe any recipient immunogenic reaction towards the canine-derived PRFs.

The released GFs are recognized as endogenous peptides that regulate both fibroblast and peripheral stem cell migration, proliferation, and differentiation, also promoting angiogenesis, crucial for the wound healing processes [1,3,31]. The inexistence of infection throughout the healing process may disclose inherent PRF antimicrobial properties, a characteristic demonstrated by other studies with human clots [39,40].

The data collected within this study demonstrate a significant difference between all the time points, reflected by the percentage of the wound area decreasing over time and a sustained healing pattern, probably supported by the constant concentration of growth factors and cytokines provided by each PRF matrix [18].

The lack of a control group with analogous lesions treated without the grafting of PRFs in this study is a limitation of this research, but this study was focused on traumatic wounds, which were difficult to reproduce since the majority of the wounds were naturally contaminated. The potential treatment of half of the wounds with PRFs and the other half with a saline solution or with a conventional method (for instance, a commercial wound ointment) would not be scientifically valid, since the PRFs’ secretome would be diffused into the control region due to its in situ degradation, as demonstrated in this study. Additionally, a control treatment group constituted by a commercial bandage or by a commercial ointment would require a higher number of cases, which was not possible, and more visits to the clinics for bandage substitution or more frequent ointment applications, which was proven to be unnecessary in PRF-therapy from the second treatment on in this preliminary study.

Moreover, a control modality with no treatment was considered unethical by the professional working group responsible for this study, consisting of surgeons, pathologists, clinical researchers, and academics with many years and extensive experience dedicated to the small animal practice area. Regarding human use of wound PRF therapy, in ten studies involving a control group, only two of them did not demonstrate better results than the control group [6]. Another study analyzing allogenic PRF-therapy in donkeys where a control group (wounds with no treatment) was used reported that complete wound closure was significantly shorter in PRF-treated wounds compared with untreated wounds (36.00 ± 1.26 days versus 28 48.50 ± 1.87 days). PRF-treated wounds healed earlier than untreated wounds, and epithelization and wound contraction were observed earlier in PRF-treated lesions [41].

The treated lesions were clinically evaluated at different but similar timepoints, considering the convenience of the owners as the factor controlling appointments. Additionally, when thinking about its performance in infected wounds, PRF therapy exhibited a preventive and effective topical antimicrobial action. Considering the actual problems associated with the existence of multi-resistant bacteria and the results attained from this study, where no topical anti-microbial or antiseptic agents were applied, PRFs may constitute a novel non-chemical agent suitable for wound healing. The authors of this study defend conventional medical therapy concomitantly with PRF therapy, according to the veterinary clinical guidelines.

## 5. Conclusions

To our best knowledge, this was the first study focusing on the clinical outcomes of canine-origin PRFs used in the management of naturally occurring wounds in compromised felines. The results of this preliminary clinical study demonstrated that canine PRF clots can be used to treat complex wounds in compromised feline recipients since the majority of the treated wounds were infected and, in some, tendon exposure was also present. Xenogenic PRFs significantly induced healthy vascularized granulation tissue in lesions with soft tissue deficit, also prompting epithelization at the injured site. This research demonstrates the safety and efficacy of PRF therapy as a biological cost-effective regenerative biomaterial that is suitable for feline wound regeneration, quickening the wound healing process. The PRF application may reduce the need for additional antimicrobial/chemical agents and even soft tissue surgeries.

## Figures and Tables

**Figure 1 vetsci-08-00213-f001:**
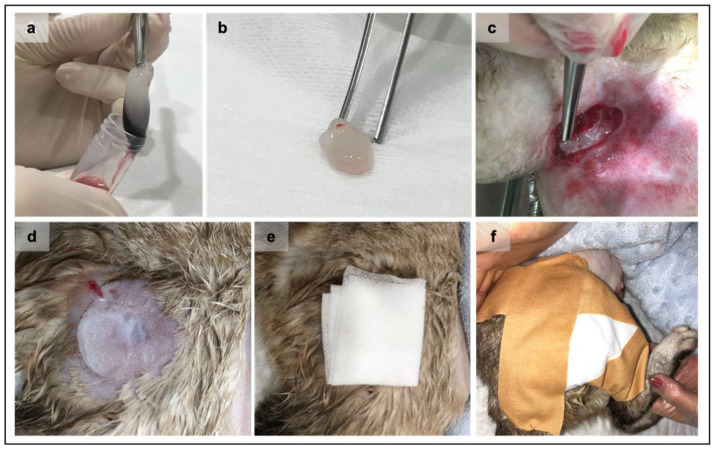
Representation of the Platelet-rich fibrin (PRF) clots grafting procedure. After the centrifugation and consolidation step, the PRF clot was gently removed from the tube (**a**) using both aseptic instruments and technique. The fibrin clot (**b**) is ready to be grafted after the removal of the red fraction. The PRF grafting procedure at wound bedding always applied an aseptic technique (**c**). Sterile petroleum jelly is applied immediately above the wound (**d**) to maintain the PRF in the wound site, followed by the placement of a paraffin gauze bandage (**e**). Finally, a closed bandage was performed (**f**).

**Figure 2 vetsci-08-00213-f002:**
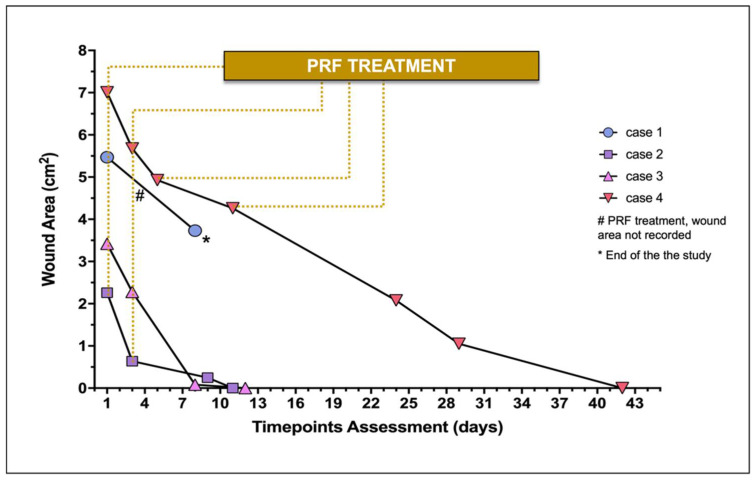
Wound area of treated wounds (*n* = 4) during the study. Individual wound evaluation and treatment (constituted by the PRF-grafting procedure) are represented.

**Figure 3 vetsci-08-00213-f003:**
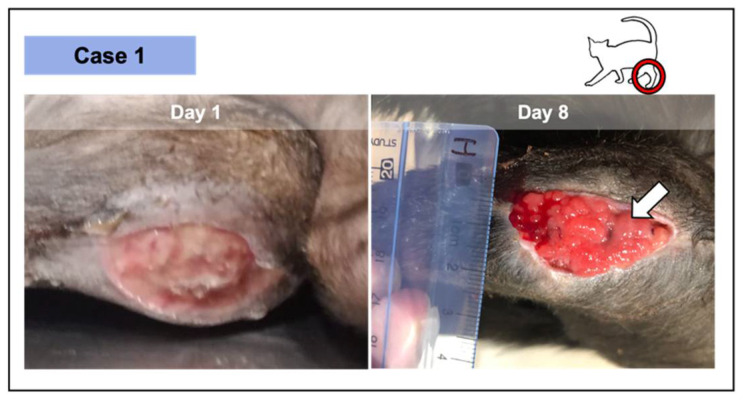
Wound from case 1, where 2 PRF grafts were implemented—at the initial appointment and at day 8. The lesion was the result of the dehiscence of the surgical repair of a metaphysis fracture at the level of the distal left femur, in this 9-month-old short-hair cat. The significant granulation tissue was verified after 8 days in an initial pale and devitalized wound (day 1). After 2 PRF treatments, epithelial tissue was perceived (white arrow) as a thin whitish layer above the vascularized area.

**Figure 4 vetsci-08-00213-f004:**
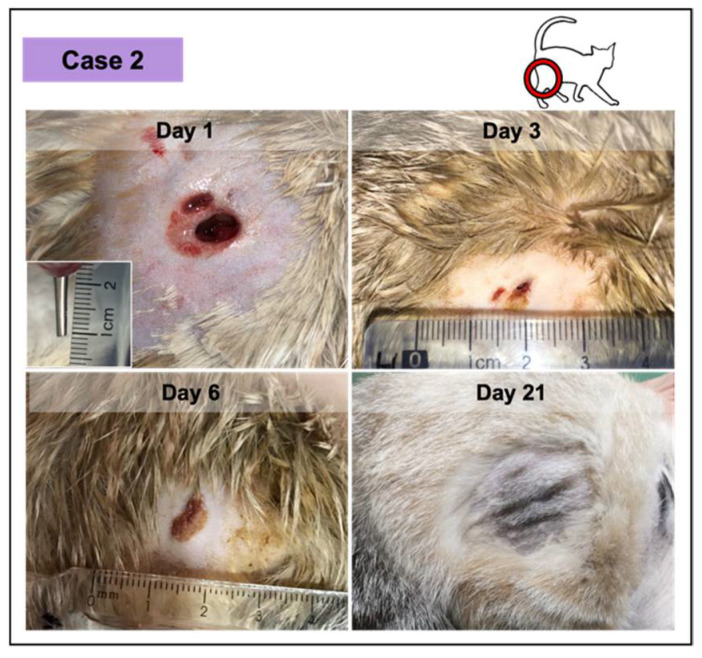
Wound healing evolution of case 2. The wound was infected, resulting from a dog bite (inset photograph demonstrating the depth at day 1). The complete wound healing was achieved at day 11. At day 21 (ten days after), the skin exhibited a normal appearance.

**Figure 5 vetsci-08-00213-f005:**
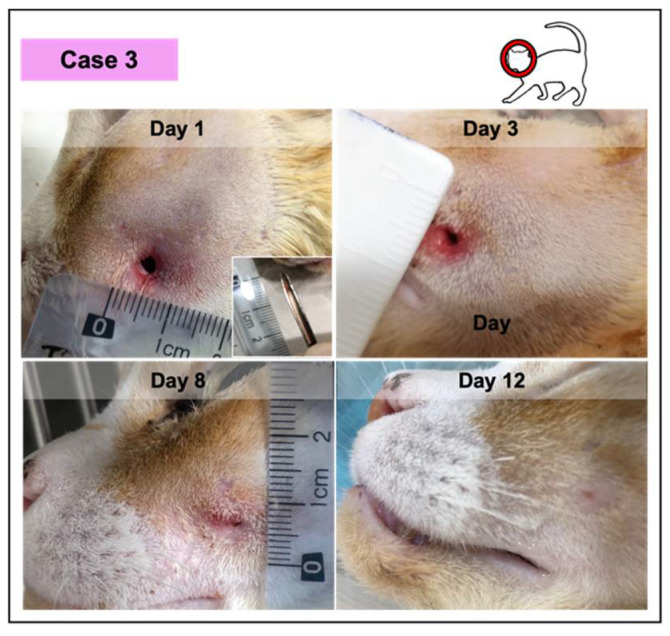
Photographic record of wound healing progression of case 3, diagnosed with an abscess with green purulent discharge. This feline was anemic and positive for the immunodeficiency virus.

**Figure 6 vetsci-08-00213-f006:**
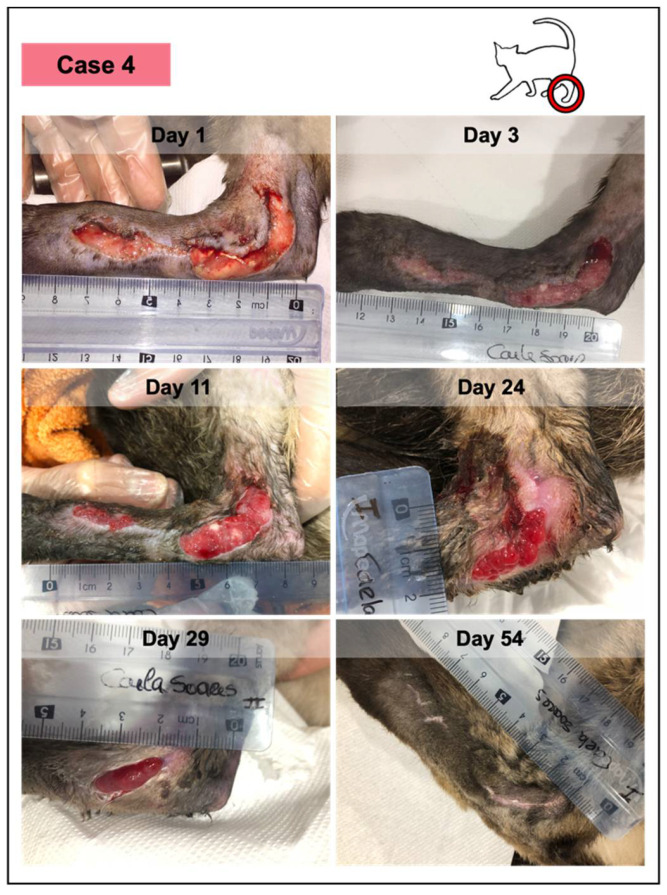
Wound healing evolution of case 4. This animal presented an extensive traumatic laceration, anemia, positivity for immunodeficiency virus, and *Bartonella* sp. According to the owner, the lesion had 48–50 h and presented a yellow purulent exudate. Medical debridement was performed under sedation (day 1), revealing a fibrinous wound bedding. Throughout the study, the patient received PRFs from different canine donors, but at different time points, considering that PRFs grafted at each timepoint were from the same donor. No adverse events were documented in any of them. The last figure (at day 54) represents the lesioned area, 12 days after the reported closure, which occurred on day 42, with complete epithelization and scar, and evident skin contraction.

**Figure 7 vetsci-08-00213-f007:**
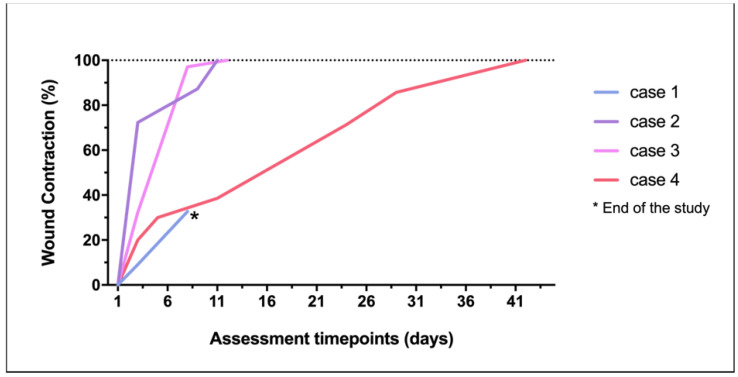
Wound contraction percentage achieved in each lesion through the study.

**Table 1 vetsci-08-00213-t001:** Characterization of canine donors.

Canine Donor	Age (years)	Sex, Reproductive Status	Breed
A	7	M, neutered	Labrador retriever
B	2	F, neutered	Beagle
C	2	M, intact	Boerboel
D	8	M, neutered	American Pit Bull Terrier

**Table 2 vetsci-08-00213-t002:** Characteristics of feline enrolled in the study.

Case	Age	Sex	Weight (kg)	Origin of the Lesion	Lesion Localization	Co-Mobilities
1	9 Mo	Male, neutered	2.600	Surgical wound dehiscence, after the repair of a metaphysis fracture	Left knee, anterior region	Polytraumatized patient (bone fracture, pancreatitis, and liver lesions) resulted from a 4th-floor fall. Submitted to orthopedic surgery after medical stabilization.
2	2 Yr	Female, neutered	3.100	Abscess due to a dog attack	Lateral region of proximal right hind limb	Fracture of the femur, a consequence of a dog attack, ventral abdominal wall laceration, hepatic lipidosis during the recovery period.
3	10 Yr	Male, neutered	4.480	Abscess resulted from a catfight	Left facial region	FIV +; mild anemia (30% hematocrit).
4	4 Yr	Male, intact	3.830	Extensive laceration with 24 h (broken glass window)	Distal hind limb, tarsus region	FIV +; moderate anemia (24% hematocrit); *Bartonella* sp. +.

Legend: FIV = feline immunodeficiency virus; Mo = months; Yr = years; + = positive.

**Table 3 vetsci-08-00213-t003:** Feline wound characterization and treatment protocol at the baseline, day 1 (D1).

Case	Time Point Day	Wound Area (cm^2^)	PRFs Applied	Donor ^a^	Wound/Study Duration	Clinical Evaluation	Adverse Effects	Considerations
1	D1	5.47	2	B	8 days ^b^	Pale color and intense fibrin deposition. Moderate exudate and thickened borders.	-	Hospitalized feline: mechanical revitalization of the lesion bed.
	D4	NP	2	A		Pink granulation tissue Moderate exudate and thickening of the borders.	Not observed	Outpatient treatment.
	D8	3.73	0 End of the study	NP		Exuberant granulation tissue and vascularization signs. Decreased exudate production. Epithelization.	Not observed	Outpatient treatment. Did not complete the study (hind limb amputation).
2	D1	2.26	1	A	11 days	Presence of infection (yellow purulent exudate). Depth wound (1.1 cm), with a muscular laceration. Erythema and oedema.	-	Hospitalized feline: surgical debridement, under surgery.
	D3	0.62	1	A		Less tumefaction. Vestigial exudate. Normal appearance of the skin.	Not observed	Hospitalized feline (hepatic lipidosis secondary to trauma).
	D6	0.25	0 Only closed bandage	NP		No exudate. Dry adherent crust. No tumefaction. Hair growing.	Not observed	The patient was discharged (hepatic recovery).
	D11	0.00	End of the study	NP		Crust detachment. Contracted skin, with no subcutaneous fibrosis.	Not observed	Reported by the owner: outpatient treatment.
3	D1	3.42	1	C	12 days	Presence of infection (green purulent exudate, with odor). Erythema and oedema.	Not observed	Hospitalized feline (under sedation).
	D3	2.28	1	C		Less tumefaction. Moderate exudate and mild erythema.	Not observed	Hospitalized feline: medical drainage, under sedation.
	D8	0.08	0 Only closed bandage	NP		Dry lesion. Evident contraction and crust formation. Hair growing.	Not observed	Hospitalized feline.
	D12	0.00	End of the study	NP		Normal skin. No subcutaneous fibrosis.	Not observed	Outpatient treatment.
4	D1	7.01	4	D	42 days	Presence of infection (purulent yellow exudate, with odor). Visualization of the ligaments/tendons. Wedge detachment. Marked edema.	Not observed	Hospitalized feline: medical debridement, under sedation.
	D3	5.68	3	D		Wound contraction and moderate exudate. Pale tissue and thickening of borders	Not observed	Hospitalized feline (under sedation).
	D5	4.93	2	D		Pink color: vascularization signs. Border with mild erythema	Not observed Uncooperative patient	Outpatient treatment.
	D11	4.85	2	A		Moderate granulation tissue: vascularization signs. Mild fibrin deposition.	Not observed Cooperative patient	Outpatient treatment. Applied PRFs from a new donor.
	D 17	NP	0 Only closed bandage	NP		Exophytic granulation tissue (PRF treatment was suspended). Mild exudate. Hair growing (local trichotomy performed).	Not observed Cooperative patient	Outpatient treatment.
	D24	2.08	0 Only closed bandage	NP		Wound contraction. Dark granulation tissue and mild exudate. Epithelialization.	Not observed Cooperative patient	Outpatient treatment.
	D29	1.05	0 Only closed bandage	NP		Wound contraction. Dry lesion Epithelialization.	Not observed	Outpatient treatment.
	D42	0.00	End of the study	NP		Normal skin. No subcutaneous fibrosis. Skin contraction.	Not observed	Outpatient treatment.

Legend: ^a^, donors are characterized in Table 1; ^b^, case 1 did not complete the study; NP, not performed.

## Data Availability

The data presented in this study are all reported in the article.

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
