# Peer review of "Canine-Origin Platelet-Rich Fibrin as an Effective Biomaterial for Wound Healing in Domestic Cats: A Preliminary Study"

_vetsci, 2021, doi:10.3390/vetsci8100213_

Round 1

Reviewer 1 Report

The study aimed to evaluate the performance of PRF of canine origin, obtained from the blood of selected donors, as a regenerative biomaterial for the treatment of wounds in felines.

The manuscript is well constructed and there are no major revisions. I have only two small considerations:

1) Was there any standardization of the PFA weight in relation to the wound size?
2) Please, it is necessary that the authors review the English of the manuscript. Check grammar and stylistic errors.

Author Response

Response to Reviewer 1

Thanks for your comments concerning our manuscript.

Considering the issue raised by Reviewer 1, questioning the “PFA weight in relation to the wound size”:

As already stated in the latest version of the manuscript, lines 122 and 123, the authors clarified and updated this version 2 (lines 122 to 124), reinforcing the idea:

“One PRF clot was applied for each 1.5 to 2 cm2 wound area, ensuring that in all the cases the wounds were completely covered by the applied PRFs. Each PRF treatment was constituted by the application of newly processed PRFs at the recipient area, assuring the contact between the PRFs platelet-rich area, and the lesion. After the PRF application, a closed bandage was applied using a sterile paraffin gauze bandage, maintaining the clot at the recipient area. “

Moreover, on day 1 (first PRF-treatment), four PRFs were applied at lesion #4, while the smallest lesion (case #2) received 1 PRF.  We have a solid experience with the PRF characteristics, mainly due to the time invested in an optimization protocol, envisioning both canine and feline PRF standardization (please see reference number 19), and is mainly for this reason that the number of PRFs allocated to each wound was done considering PRFs’ characteristics (strictly documented in table 3). Moreover, the authors recognize and agree with the recommendation raised by the Reviewer: the first version of our manuscript already described the methodology applied (regarding the number of PRFs grafted), but this was again reinforced in the second version, making the article more objective.

Thank you for your appreciation.

Reviewer 2 Report

The study is interesting. If it is well planned, well explained and well discussed.
However, the design should have included a control group in which an already used conventional treatment will be applied in order to be able to compare it with the application of PRF.
The paper describes the use of heterologous PRF of canine origin in wound management in cats. The wounds included in the study are very heterogeneous, both in terms of local conditions and characteristics as well as individual variability. This makes comparison of the results complex and somewhat inconsistent. One should not speak of evolutionary averages with only 3 completed cases. A discussion of each case could be made individually, but I think not as a whole.  The sample size is very small (3 patients).
The paper provides useful information on a novel technique in the feline species, but should make clear its limitations, objectives and approach from the outset. The title should already reflect these constraints. It would be advisable to make it clear in the title that this is a description of several clinical cases in the form of a clinical or preliminary study. The current wording of the title does not make this fact clear, the reader expects a detailed study of the effects and uses of heterologous PRF in the cat and this is not the aim of the paper.

Author Response

The authors would like to acknowledge all the suggestions raised by the reviewer, that were made in the new version along with the manuscript (version 2).

The authors recognize the proposal raised by the reviewer, and the title was restructured.

The authors consider that the limitations of this novel technique (the PRF-grafting technique) are copiously addressed and supported with recent scientific literature, when xenogenic hemoderivative clinical use is applied, at the “Discussion” section.

We understand that the discussion of individual cases could be more suitable, and, in fact, it started to be described as suggested by the Reviewer. Nevertheless, the research team found a repetitive text, once these cases have without exception the same healing/ evolutionary phenomena, as demonstrated by the photographic data.

The authors also consider that the presentation of evolutionary averages represents a scientific demonstration, reflecting the wound area reduction along the PRF-therapy. Nevertheless, the authors altered the mentions to wound area averages, and individual areas are now specified.

The authors also agree and assume that the sample size studied is reduced, but this study is based on traumatic wounds, being limited to the patients presented in the clinics during a specific time period, and considering cases where surgical reconstruction was not the first option.

Finally, the team assumes “The lack of a control group with analogous lesions treated without the grafting of PRFs in this study is a limitation of this research” (Discussion section, line 373). Nevertheless, this study was focused on demonstrating the clinical performance and effectiveness of PRF-therapy in a real clinical scenario, in real soft tissue injuries, documenting its clinical safety and outcomes.

Additionally, spontaneous wounds especially the ones presented in this paper are difficult and even impossible to experimentally reproduce since the majority of the wounds were naturally contaminated. Treating half of the wound with PRF and the other half with a saline solution or with a conventional method (for instance, a wound ointment) would not be valid, since PRFs’ secretome would certainly be diffused into the “control region” due to its in situ degradation, demonstrated in this and in a previous study (reference 20).  

On the other hand, the creation of artificial wounds versus a control modality with no treatment was considered unethical by the professional working group responsible for this study, constituted by clinical researchers and academics dedicated to the small animal practice area. Moreover, the most experienced member of our group has more than 20 years as a veterinary surgeon, and particularly in wound healing (Professor Dr. Isabel Dias), and the healing phenomena attained by the PRF-grafting technique is, according to her, incomparable to the conventional treatments such as commercial healing ointments. Also, the scientific literature consulted reinforces that this therapy is most of the time more efficient than conventional therapy (as presented in the Discussion section, supported by scientific bibliography).
